# FRP Pedestrian Bridges—Analysis of Different Infill Configurations

**Lucija Stepinac** [1,*], **Ana Skender** [2] , **Domagoj Damjanović** [2] and **Josip Galić** [1,*]

1 Faculty of Architecture, University of Zagreb, Andrije Kačića Miošića 26, 10000 Zagreb, Croatia
2 Faculty of Civil Engineering, University of Zagreb, Andrije Kačića Miošića 26, 10000 Zagreb, Croatia; ana.skender@grad.unizg.hr (A.S.); domagoj.damjanovic@grad.unizg.hr (D.D.)
* Correspondence: lstepinac@arhitekt.hr (L.S.); jgalic@arhitekt.hr (J.G.)

**Abstract:** The main aim of this study is to analyze fiber-reinforced polymer (FRP) bridge decks according to their material, cross-section, and shape geometry. Infill cell configurations of the decks (rectangular, triangular, trapezoidal, and honeycomb) were tested based on the FRP cell units available in the market. A comparison was made for each cell configuration in flat and curved bridge shapes. Another comparison was made between the material properties. Each model was computed for a composite layup material and a quasi-isotropic material. The quasi-isotropic material represents chopped fibers within a matrix. FE (finite element) analysis was performed on a total of 24 models using Abaqus software. The results show that the bridge shape geometry and infill configuration play an important role in increasing the stiffness, more so than improving the material properties. The arch shape of the bridge deck with quasi-isotropic material and chopped fibers was compared to the cross-ply laminate material in a flat bridge deck. The results show that the arch shape of the bridge deck contributed to the overall stiffness by reducing the deformation by an average of 30–40%. The results of this preliminary study will provide a basis for future research into form finding and laboratory testing.

**Keywords:** FRP deck; pedestrian bridges; pultruded deck; sandwich deck; cell configuration; laminate; quasi-isotropic; geometry optimization; 3D printing

## 1. Introduction

Henry Ford, a great innovator in the auto industry, introduced FRP material to the world under the motto "Ten times stronger than steel" [1]. However, even though it is not a novelty in developed countries, it is still not widely used in developing countries.

FRP is a two-component material whose volume consists of 30–70% fiber, with 50% of its total weight incorporated into a polymer matrix. With regard to the mechanical characteristics of FRP, fibers are the ones in charge of carrying the load and providing strength, stiffness, and thermal stability. The matrix wraps the fibers and protects them during the production process and during the exploitation time, ensuring an even load distribution to each fiber. It is also crucial in providing composite durability [2].

At this point, the market is full of various fibers. Glass fibers (GFRP) are the most common choice for investors, architects, and structural engineers due to their good mechanical and physical properties and low price compared to carbon fibers (CFRP), aramid fibers (AFRP), and basalt fibers (BFRP).

There are many applications of FRP in bridge engineering, mostly with pedestrian bridges. FRP material is a good alternative to traditional materials that will extend the life of the structure or enable the usage of pedestrian bridges for a longer period. The replacement of parts in the existing bridges, such as concrete decks, with FRP decks, or even replacing the whole structure, can be performed without significantly disturbing the structure. Composites excel because of their high modulus of elasticity, ultimate load capacity, and low density, making them ideal for strengthening the existing girders and

decks and using them in a new, fully composite or hybrid bridge structure [3]. Some of this material's other beneficial properties include rapid construction, anticorrosive properties, water resistance, pleasant appearance, lasting color, overloading resistance, good dynamic performance [4], and the potential to build bridges with a greater span [5]. In addition, research by Mara [6] shows that less energy is needed for the production and maintenance of FRP bridges than for construction based on other materials.

FRP is promising in terms of its durability, but its high price makes it difficult for it to compete with traditional materials. Insufficient building codes and poor knowledge of the benefits of anisotropic materials are the reasons why architects and structural engineers often choose traditional materials over FRP.

The research from M. Gunaydin et al. [7] on the Halgavor bridge details the use of three different materials: steel, GFRP, and a steel–GFRP combination. The dynamic analysis results are positive for when only GFRP is used in a static representation of the Halgavor suspended bridge. Comparing the total weight, GFRP has only half of the weight of the steel bridge and almost a fifth of the weight of the concrete bridge. Structural and physical appearances are of great value, and architectural design should be taken into consideration more often when designing with FRP. Each new material must go through a process of finding the optimal structural system and the shape that suits it best, but that involves copying other traditional materials at the beginning. There are numerous examples of when FRP decks are used on steel and concrete girders [8–11]. FRP composites make a good alternative to traditional plate systems due to their strength-to-weight ratio and their pleasant appearance [12].

In this research, several FRP decks were numerically investigated to show how geometry modification can influence the global behavior of the panels—in this case, FRP bridges.

The main aim of this research is to see how the inclination and shape of the bridge, its infill geometry, and its material can affect stiffness. We assume, in our models, that the type of material is known. Twenty-four models were made. Twelve of them have been modeled with an anisotropic GFRP material and a layup orientation in two orthogonal directions (0/90/0/90). Another 12 models have quasi-isotropic material properties, including apropos matrix mixes with chopped fibers. Quasi-isotropic material will result in a higher deformation and a lower load-bearing capacity due to its lower modulus of elasticity and strength. A curved bridge geometry will result in lower deformation and higher stiffness. The main aim of this preliminary research is to quantify the level of this reduction according to material and geometry. The reason for the use of quasi-isotropic material (even though its modulus of elasticity is much lower than laminated alternatives) is because of the application of FRP in more advanced production processes, such as 3D printing. The configuration of infill cells in a 3D printed bridge is used to define a self-standing bridge deck. No additional girders under the deck are required, as suggested by [13], while 2.7 m is the standard span length for currently available sandwich panels on the market. The "sandwich deck" is the main load-bearing structure and is supported on two edges. Smart formworks can also be used, as well as 3D printing. Thus, this research could be applicable to laminate, too.

## 2. FRP Bridge Decks

### 2.1. Structural Systems

After defining the material properties, it is necessary to define a cross-section of the element. The infill geometry is based on established plate systems that can be found on the market. Four of them have longitudinally oriented cells with different web inclinations, and two have orthogonally oriented cells, known as sandwich panels. Designing a bridge in the form of an arch enabled us to quantify an increase in stiffness and, consequently, a deflection reduction.

Plate bridge deck systems have not changed much since 1991 [14]. The most common systems are sandwich panels and pultruded profiles that are adhesively connected. The technology of pultrusion is the most commonly used in the FRP production process, and

for those reasons, the most common plate systems available on the market are EZSpan (Atlantic Research, Gainesville, VA, USA), Superdeck (Creative Pultrusions, Alum Bank, PA, USA), DuraSpan (Martin Marietta Materials, Raleigh, NC, USA), and Strongwell. Sandwich panels consist of top and bottom flanges that carry the load and a low-weight infill that connects them and transmits the load but does not contribute to the panel's stiffness [15].

In a recent review paper [16], a summary of FRP pultruded deck and sandwich panel failure modes has been provided [13,17–19]. It can be concluded that FRP decks demonstrate linear elastic behavior up until failure. Some pseudo-ductility could be achieved through the structural system and cell configuration [16].

An example of a temporary bridge structure [20] shows that after eight years of service, some visible damages were: flange crushing, longitudinal cracks, visible fibers due to top-surface blooming, and some local damages. Despite these flaws, structural stability was not yet jeopardized, and all of the damages were easily repaired, for the connections stayed in perfect condition.

It has been reported [21] that for triangular cell configurations in a pultruded deck (EZ Span and Asset deck), only around 20% of the material compression strength is utilized. This is caused by local buckling and delamination between the web and flanges, which are the symptoms of deck failure. Local deformation or cracking of the top surface or wear surface will occur under patch load, well before the compression strength limit of the material is reached. For this reason, many parameters of FRP decks should be considered during the design to obtain a more optimal, cost-acceptable, attractive, and sustainable solution.

The pultruded plate systems that are currently available on the market vary in depth from 80 to 225 mm depending on the production process (see Table 1). Sandwich panels, on the other hand, are more flexible in terms of production dimensions, and can also be inclined. An FRP composite structural system with FRP sandwich decking laying on a U-box girder with a bridge span of 15–25 m has recently been proposed and examined in Poland [22,23].

**Table 1.** Physical properties of plate systems from adhesively bonded pultruded profiles [6].

| System | Production Process | Thickness (mm) | Weight per m$^2$ (kN/m$^2$) |
|---|---|---|---|
| Superdeck | Pultrusion | 203 | 1.0 |
| DuraSpan | Pultrusion | 195 | 1.05 |
| EZSpan | Pultrusion | 216 | 0.96 |
| Strongwell | Pultrusion | 170 | - |

The FRP's load-bearing shell structure can be flat or curved. A shell in arch has a higher stiffness and a smaller deformation. The problem with forming the elements in an arch is that additional material is required at the area where the bending moment is significant. A bridge made in cooperation with the FiberCore company [24] is a good example of where each bridge structure's design fulfills the required, ultimate serviceability limit states for every member within the allocated costs. Lately, optimization with additional software, such as Grasshopper, Karamba, and Kangaroo, has been of great help [25].

Many aspects of FRP decks affect the final results, such as material (e.g., GFRP or CFRP), the configuration and inclination angle of the internal cavities, and thicknesses, etc. It is shown [9,26] that a triangular configuration in pultruded decks has a higher in-plane shear stiffness ($G_{xz}$) compared to a rectangular configuration. In sandwich panels, the density of the cavities will influence stiffness more than geometry. Sandwich panels can be filled with polymeric foam in order to increase stiffness [6]. Sandwich decks usually have sufficient shear stiffness to transfer these strains, whereas pultruded FRP decks have lower shear stiffness.

### 2.2. Production

Several technological processes have been used to produce FRP elements depending on the required shape and structural parameters. One of the first was the hand-lamination process, which is fully manual, as indicated by the name. The number of products that can be made is limited, but the form and shape of the products is limitless, depending on the formwork. Pultrusion is when fibers soaked in a matrix go through a heated mold where the element is shaped. This is a fully automated process, and after the element leaves the mold, it is cut at the required length. Although the size of continuous cross-section limits the process, the quality of the produced members is of the highest standard among all currently established production processes. It is also one of the most economical processes.

Technology is leading us to smarter and more cost-effective solutions. An important recent innovation in the production of curved-plate elements has been the use of smart formworks, especially when these are used not for a series of the same product, but for the creation of a unique product. At the moment, these formworks are well known in the production of concrete [27] and glass panels, but it could also be applicable to FRP. Greater freedom of architectural design is one of the most important advantages that this process offers.

Another growing industry in pedestrian bridge production is 3D printing technology and production assisted by robotic arms. There are few examples of 3D-printed steel bridges, but also some promising projects that involve printing with FRP. An example of a 3D-printed pedestrian bridge can be found in China from 2020 [28], which has total length-width dimensions of 15.5 m × 3.8 m (see Figure 1). The total load capacity is estimated to be 250 kg/m$^2$, and it has an expected lifetime of 30 years. The material is ASA (acrylonitrile styrene acrylate), reinforced with glass fibers. The total weight of the bridge is 5.8 t, and it has a fiber content of 12.5%. Total production took 30 days on a 3D printer, which had the capacity to print 8 kg/h and a maximum volume of 24 m × 4 m × 1.5 m. The main problem is that 3D printing is still a relatively slow process compared to the construction of bridges with traditional materials and techniques.

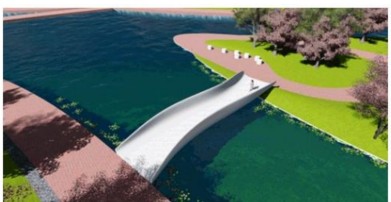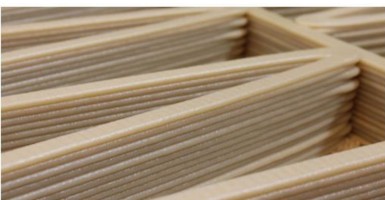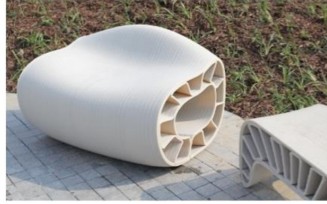

**Figure 1.** First 3D-printed bridge in China, Shanghai (**left**); 3D printing infill pattern (**middle**); 3D-printed cross section (**right**) [28].

The pultrusion production process limits the FRP bridge deck to unidirectional decks with a straight and constant depth. Unlike pultruded decks, sandwich decks can be produced with varying depths and larger sizes. Sandwich panels include two stiff and strong face plates adhesively connected to a core [16]. The core material can vary in configuration (e.g., honeycomb, triangular, sinusoidal) and density. A denser raster will result in greater stiffness. The optimization of the core material within the deck could be obtained if the denser core would appear in areas of higher stresses. Moreover, novel materials for higher stiffness and strength could be utilized, resulting in a more elegant structure [29,30].

The most cost-effective production process will likely govern the final bridge design. In this study, longer life cost–benefit is considered. The totality of the labor could be automatized without sacrificing another aspect of the project, such as time and money saving, or the appearance of the structure. Production processes assisted by robotic arms, 3D printers, and smart formworks are some of the possible solutions. These are costly in the beginning but easily adaptable to various project solutions. More sophisticated

technologies would allow variations in slope, depth, and 3D core configuration to be integrated into the FRP bridge deck. Thus, additional material could be added in places where higher local stiffness is needed. A subject for future research will be the performance of 3D cell configurations [31].

*2.3. Connections*

The serviceability limit state will most likely govern the design in the structural analysis unless there are mechanical connections. In this case, connection location becomes critical, rather than the element itself. In adhesively connected elements, the load is transferred uniformly and, for that reason, this feature is considered to be more appropriate in FRP structures.

During the design of such FRP plate systems, the main purpose of these connections is to transfer bending moment and shear force between two systems without significant deflections. Laboratory tests showed that the collapse of the component would occur before a connection failure occurs. Laboratory examinations for more significant projects, where unreasonably high partial safety factors are applied, lead to oversized structures [32]. The study and testing of FRP material [33] will reduce the time required for laboratory testing and facilitate the design of structures with this material.

The connections and adhesive bonds between elements in pultruded decks can significantly influence the global stiffness [26,34]. The force is transferred more uniformly when adhesive connections are used, making them more appropriate for FRP structures. Mechanical connectors are mandatory when an element exceeds limitations in the production process or during transportation.

According to the Eurocomp standard and manual [35], there are three types of connections: connections that should last a whole lifetime and whose collapse would be devastating for the whole structure; connections whose collapse would affect the structure locally and would not have a serious impact on the structure as a whole; and non-structural connections which basically connect secondary elements, such as a fence.

## 3. Materials and Methods

In the previous section, a general overview of existing research on FRP decks was presented. Based on the literature review on FRP deck systems, two general groups of products can be distinguished: panels made of adhesively pultruded profiles and sandwich panels [36]. Models of bridge decks consist of three adhesively bonded components (the top surface, web, and bottom surface).

In this study, a bridge deck with two hinge supports and a span of 5 m was used. The depth of all of the decks was uniform at 200 mm, while the width was 1.8 m. Uniform wall thicknesses of 6 mm were used in all cross-sections (four layers of 1.5 mm in two orthogonal directions (0/90/0/90)). The main aim of this study was to have a similar depth and mass for all decks. Thus, the stiffness would be primarily influenced by the cell configuration (Figure 2).

Different infill geometries and web inclinations were analyzed using Abaqus software. Panel depth was taken as 200 mm according to the available standard panel systems (Figure 3): (a,b) EZ-Span (49.91 kg/m$^2$), (c,d) Superdeck (50.48 kg/m$^2$), (e,f) DuraSpan (50.74 kg/m$^2$), and (g,h) Strongwell (41.08 kg/m$^2$), which are made of pultruded profiles, and sandwich panels: (i,j) with a profiled triangular infill (74.84 kg/m$^2$) and (k,l) honeycomb sandwich panels (54.19 kg/m$^2$). The geometry of the deck cross-sections (rectangular, triangular, trapezoidal, and honeycomb) was taken from the different FRP cell units available in the market. Each plate system weighs approximately 50 kg/m$^2$. Future research will be performed for 3D cell configuration [37].

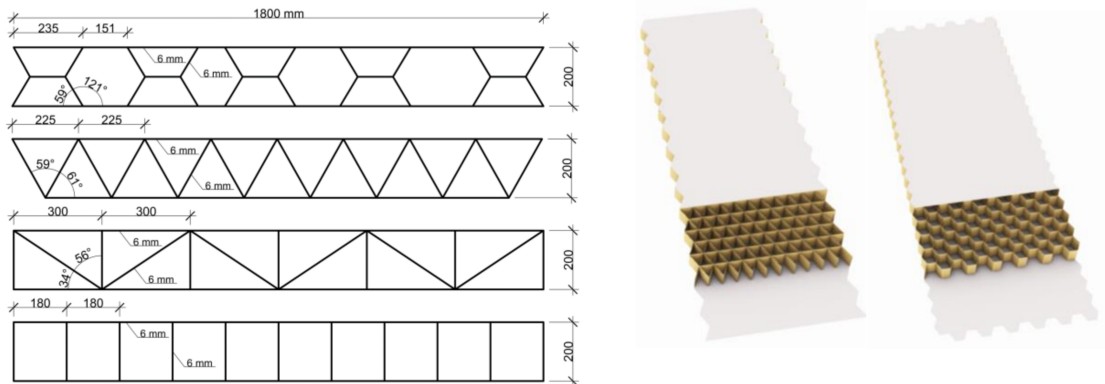

**Figure 2.** Cross-section of pultruded deck panels in mm (**left**); sandwich panels (**right**).

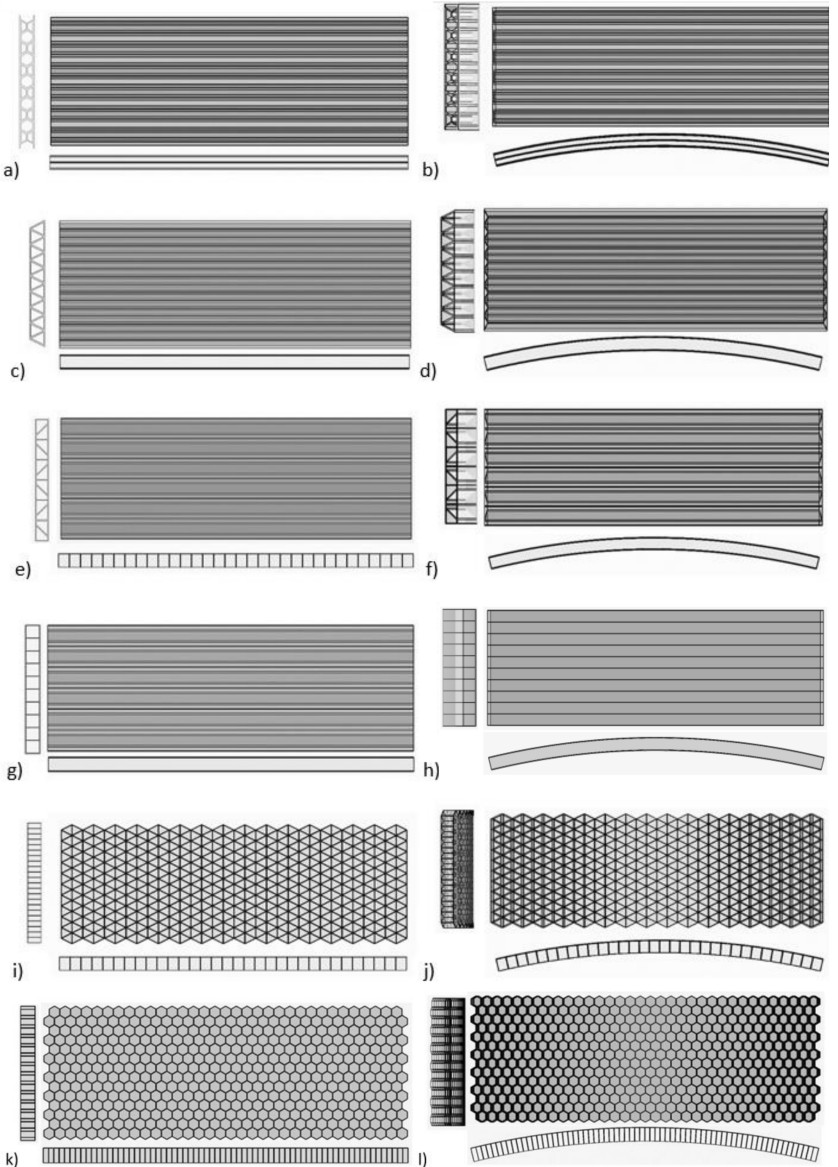

**Figure 3.** GFRP plate, adhesively bonded pultruded profiles: (**a**) Superdeck; (**b**) Superdeck in an arch; (**c**) EZSpan; (**d**) EZSpan in an arch; (**e**) DuraSpan; (**f**) DuraSpan in an arch; (**g**) Strongwell; (**h**) Strongwell in an arch. Sandwich panels: (**i**) triangular-shaped infill; (**j**) triangular-shaped infill in an arch; (**k**) honeycomb-shaped infill; (**l**) honeycomb-shaped infill in an arch.

The geometry was defined using the Rhinoceros software package. The serviceability load combination was applied to the top of each deck according to the British code [38]. A distributed live load of 5 kN/m² and a dead load of 1 kN/m² was applied (in case of wearing, the surface is applied).

In this study, a simple arch with a height of 30 cm at the middle of the span was developed. With innovative production processes, such as 3D printing, more design freedom is possible. Optimal structures, based on internal forces, could be created. The selection of bridge geometry would not be determined by the location of the bridge.

In addition, a comparison was made between a composite fiber layup (0/90/0/90) material and a quasi-isotropic material. The composite layup consists of four orthogonally positioned fiber layers, each 1.5 mm thick, with their material properties shown in Table 2. On the other hand, one modulus of elasticity is defined for the quasi-isotropic material. The modulus of elasticity for composites reinforced with glass fibers can vary from 5 GPa to 50 GPa [39]. For a resin reinforced with 40% of chopped strand mat by weight, a modulus of elasticity of 10 GPa was chosen (see Figure 4). Three-dimensional models were imported from Rhinoceros software into Abaqus software as surface shell parts. Tie constraints connected the top and bottom surfaces to infill surfaces, thus simulating the rigid behavior of all composite panel board instances. The pedestrian bridge was analyzed as a clamped supported beam and a clamped arch. Finite element modeling was performed with quadratic mesh size elements of 50 mm × 50 mm.

**Table 2.** Composite layup material properties for glass fiber epoxy resin [40].

| $E_1$ (MPa) | $E_2$ (MPa) | $\nu_{12}$ | $G_{12}$ (MPa) | $G_{13}$ (MPa) | $G_{23}$ (MPa) | $\rho$ (kg/m³) |
|---|---|---|---|---|---|---|
| 34,412 | 6531 | 0.217 | 2433 | 1698 | 2433 | 2000 |

$E_1$: Young's modulus in longitudinal (fiber) direction. $E_2$: Young's modulus in transverse. $\nu_{12}$: Minor Poisson ratio. $G_{12}$: In-plane shear modulus. $G_{13}$: Out of plane shear modulus. $G_{23}$: Out of plane shear modulus. $\rho$: Density.

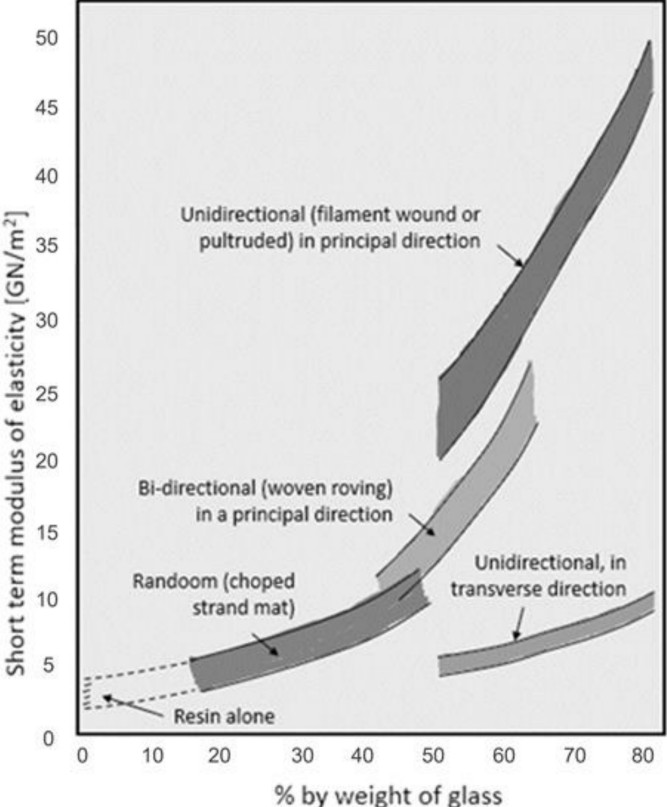

**Figure 4.** Modulus of elasticity for glass fiber reinforced polymers [39].

## 4. Results

The most effective cell unit configuration was determined based on the serviceability design criteria as governed by deflections. According to the British code [38], a deformation limitation of L/300 (16.6 mm) was compared to the maximum deformation at the middle of the span. To obtain the deformations, the complex numerical modeling of several FRP decks (see Figure 2) was performed in Abaqus. The results are shown in Figures 5–10.

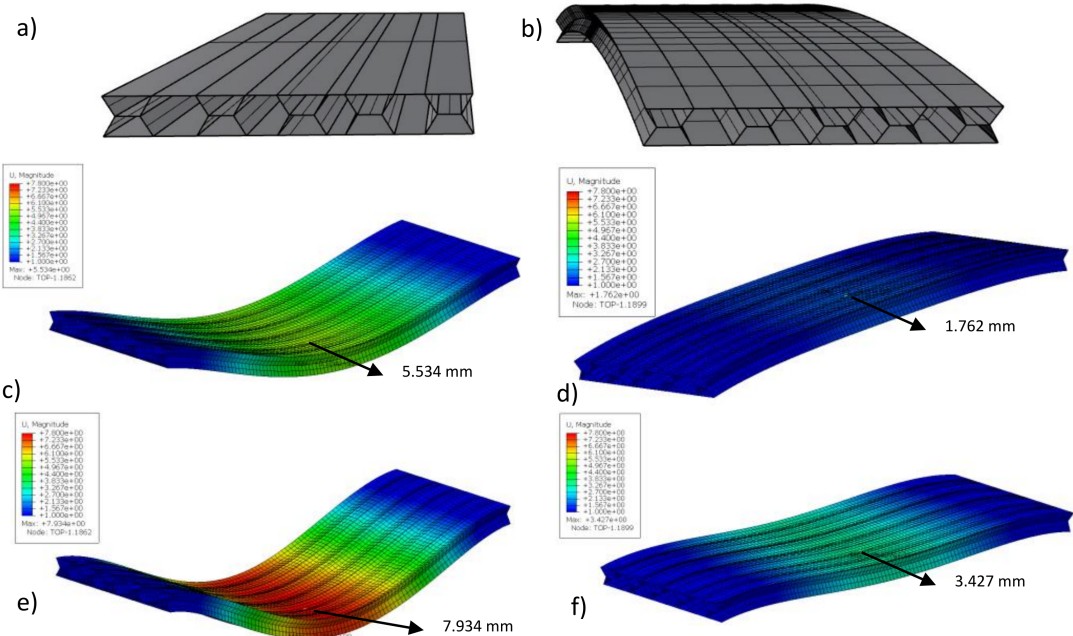

**Figure 5.** Geometry of: (**a**) Superdeck and (**b**) Superdeck in an arch. Deformation for: (**c**) Superdeck (laminate); (**d**) Superdeck (laminate) in an arch; (**e**) Superdeck (quasi-isotropic); (**f**) Superdeck (quasi-isotropic) in an arch.

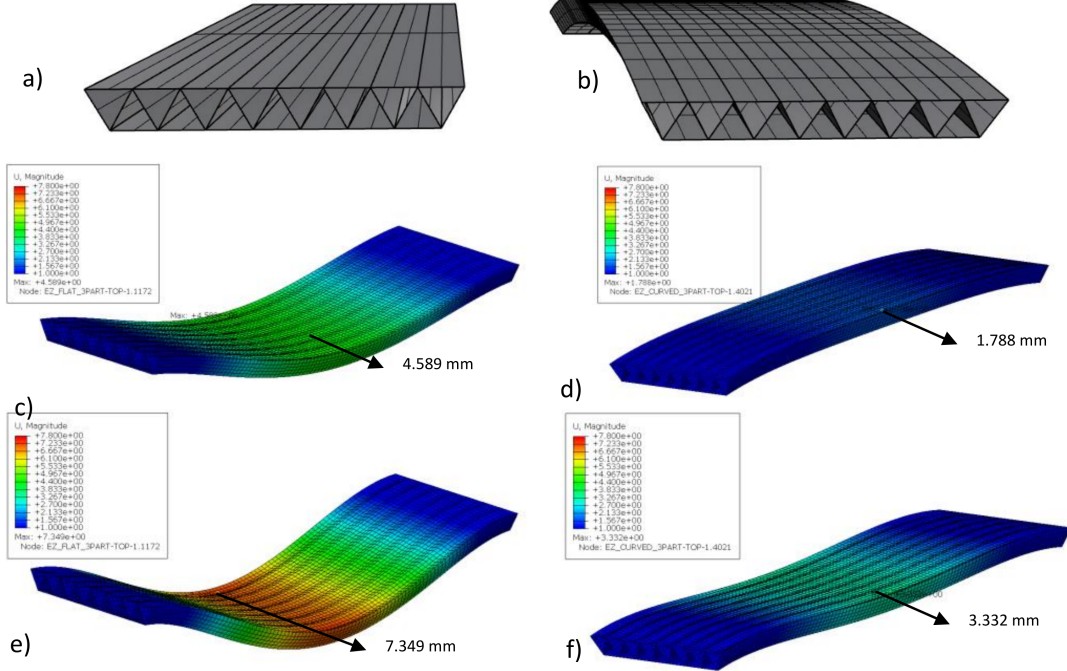

**Figure 6.** Geometry of: (**a**) EZSpan and (**b**) EZSpan in an arch. Deformation for: (**c**) EZSpan (laminate); (**d**) EZspan (laminate) in an arch; (**e**) EZspan (quasi-isotropic); (**f**) EZspan (quasi-isotropic) in an arch.

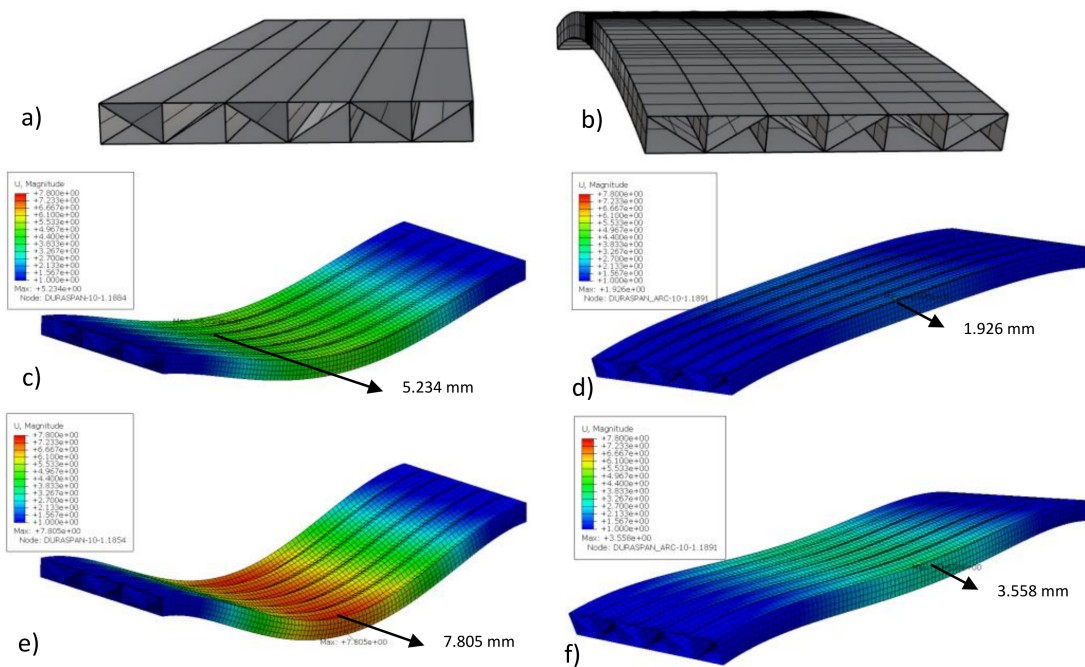

**Figure 7.** Geometry of: (**a**) Duraspan and (**b**) Duraspan in an arch. Deformation for: (**c**) Duraspan (laminate); (**d**) Duraspan (laminate) in an arch; (**e**) Duraspan (quasi-isotropic); (**f**) Duraspan (quasi-isotropic) in an arch.

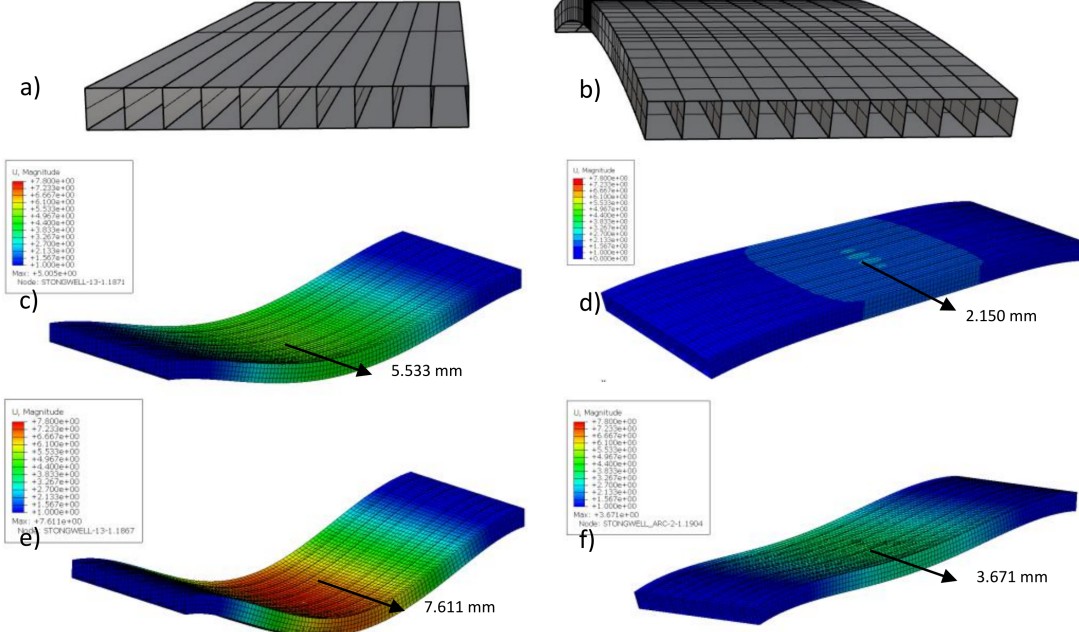

**Figure 8.** Geometry of: (**a**) Strongwell and (**b**) Strongwell in an arch. Deformation for: (**c**) Strongwell (laminate); (**d**) Strongwell (laminate) in an arch; (**e**) Strongwell (quasi-isotropic); (**f**) Strongwell (quasi-isotropic) in an arch.

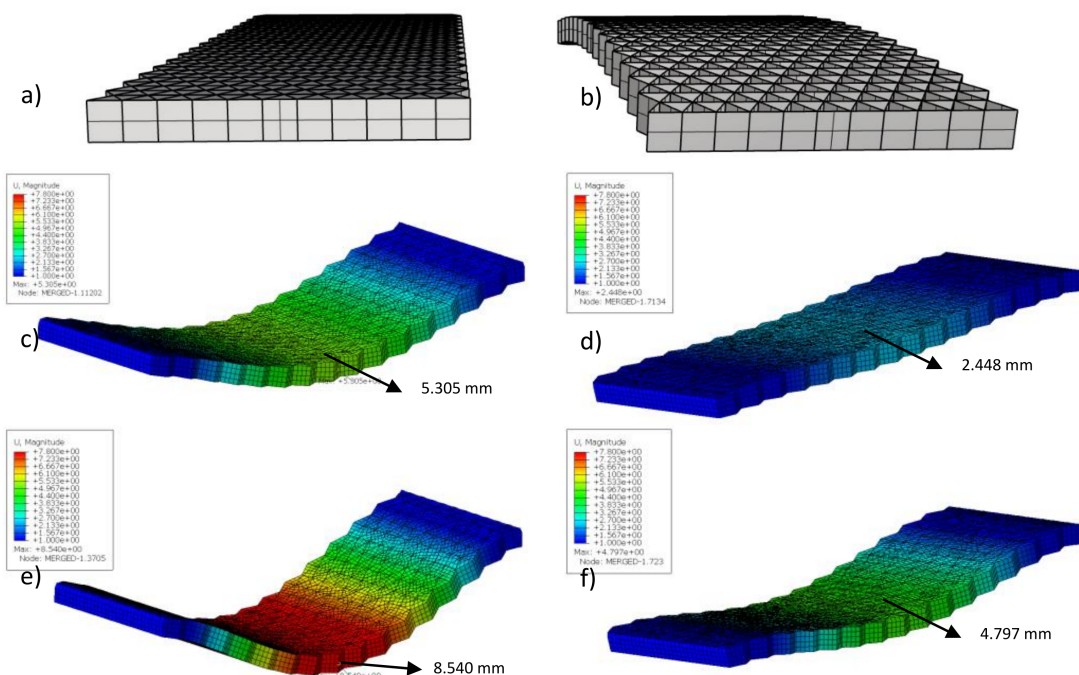

**Figure 9.** Geometry of sandwich panel with: (**a**) triangular cell configuration and (**b**) triangular cell configuration in an arch. Deformation for sandwich panel with: (**c**) triangular cell configuration (laminate); (**d**) triangular cell configuration (laminate) in an arch; (**e**) triangular cell configuration (quasi-isotropic); (**f**) triangular cell configuration (quasi-isotropic) in an arch.

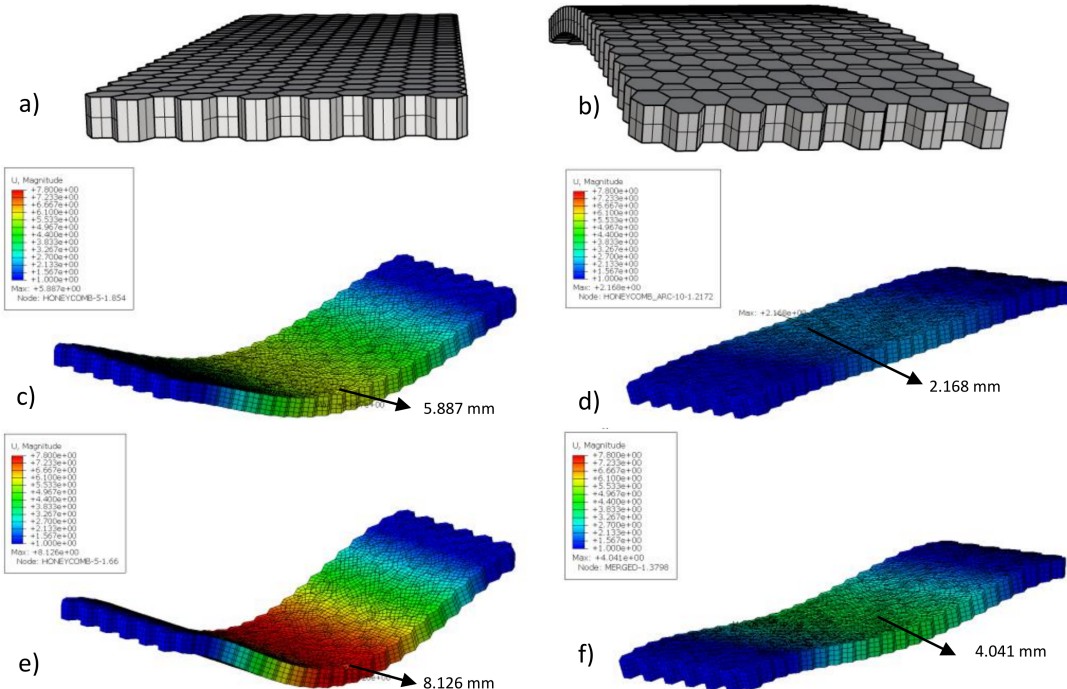

**Figure 10.** Geometry of sandwich panel with: (**a**) honeycomb cell configuration and (**b**) honeycomb cell configuration in an arch. Deformation for sandwich panel with: (**c**) honeycomb cell configuration (laminate); (**d**) honeycomb cell configuration (laminate) in an arch; (**e**) honeycomb cell configuration (quasi-isotropic); (**f**) honeycomb cell configuration (quasi-isotropic) in an arch.

The comparison of the results is shown in Figure 11. Since the serviceability limit states govern the design of these types of bridges, the displacements are more relevant for the analysis.

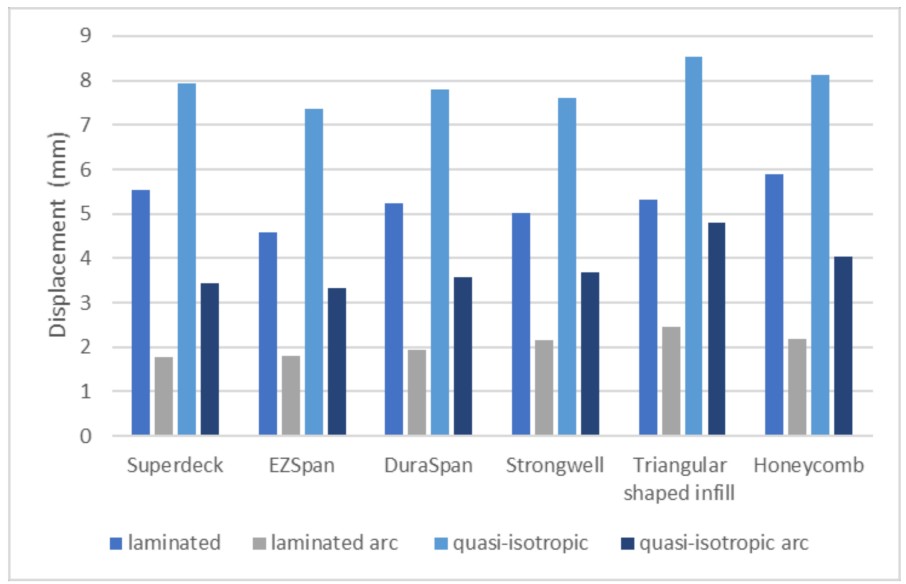

**Figure 11.** Results for displacements.

The study showed that the lowest deflections of about 1.75 mm were obtained in the arched Superdeck (which has a hexagonal cell configuration) and EZ Span (with a triangular cell configuration) bridge decks. The decks made of pultruded profiles were stiffer overall than the sandwich panels. In previous research by [41], the authors found that Superdeck (hexagonal cell) and ASSET (triangular cell, similar to EZ Span) were stiffer than others, which is in accordance with the results of this study.

Bilim et al. [41] investigated the influence of the additional concrete layer on the top face and concluded that the displacements of the bridge deck are reduced by an additional 60%. Several studies have been conducted showing an increase in local stiffness after the application of the wear layer [11,21]. In this study, no additional wear layer was applied that could influence the stiffness, but it could be part of future research.

The midspan displacement values for decks with a quasi-isotropic material exceed the midspan displacement values for decks with the laminate layup (0/90/0/90) by 32% on average for flat decks and by 46% on average for arched decks. Thus, the calculated mass per m$^2$ and midspan displacement values are given in Table 3 for arched and flat decks with the laminate layup. For example, the arched Superdeck system has a smaller deflection than the Strongwell system (by 18%), but a higher value in mass per m$^2$ (by 23%) than the Strongwell system.

**Table 3.** Results for composite layup (0/90/0/90) plate systems.

| Plate System | Flat | Arch | Mass Per m$^2$ |
|---|---|---|---|
| | Displacement (mm) | | (kg/m$^2$) |
| Superdeck (laminated) | 5.53 | 1.76 | 50.48 |
| EZSpan (laminated) | 4.59 | 1.79 | 49.91 |
| DuraSpan (laminated) | 5.23 | 1.93 | 50.74 |
| Strongwell (laminated) | 5.01 | 2.15 | 41.08 |
| Triangular (laminated) | 5.31 | 2.45 | 74.87 |
| Honeycomb (laminated) | 5.89 | 2.17 | 54.19 |

The results show that the maximum deflection occurred in the flat bridge with quasi-isotropic material properties, reaching 8.5 mm in the sandwich panel with triangular infill configuration.

The main conclusion can be drawn by comparing the calculated displacements in the flat laminated decks with the quasi-isotropic decks in an arch. Even though the quasi-isotropic material with the chopped strands has lower material properties than the cross-ply laminate, the arch shape of the bridge deck contributed to the overall stiffness by reducing the deformation by around 30–40%.

## 5. Discussion and Conclusions

FRP as a material is still under investigation and is waiting to fulfill its potential in the construction industry. All aspects, from geometry, to material parameters, to connections, to its production, are important for a successful bridge design.

In this study, finite element modeling of FRP bridge decks made of pultruded profiles and sandwich panels was performed in Abaqus. Typical infill geometries (rectangular, triangular, trapezoidal, and honeycomb) were used, with two different materials for each infill, namely a quasi-isotropic material and a cross-ply laminate. The objective of this research was to analyze the effect of the arch shape of the bridge deck on the overall stiffness when compared to a flat bridge deck. An increase in deflection was observed for the quasi-isotropic material in all of the bridge decks considered here when compared to the laminated decks. On the other hand, there is also an obvious reduction in deformation for the plates formed in an arch. Nevertheless, there is a 30–40% stiffness increase in curved bridges with quasi-isotropic material properties compared to laminate bridge decks in a flat shape. In pultruded decks, hexagonal and triangular infill configurations show smaller deformations than rectangular, while in sandwich panels, honeycomb is the best configuration. When comparing pultruded decks and sandwich panels under a uniform load, pultruded profiles with longitudinal cavities are stiffer. There are research possibilities for finding new infill configurations, especially in 3D infill geometries that could vary in density and cross-sectional height. The results justify the use of 3D printing technology and polymeric filament that allow variations in the shape and infill cell configuration of the bridge structure.

Form finding could reduce the material cost of retaining the structural integrity. In future research, the investigation of and laboratory tests for material properties are mandatory. Bridge self-weight plays an important role in decision making. The main idea is the definition of curved shapes and tridimensional infill configurations to find out which would give the lightest possible bridge structures.

Analytical approaches through software such as Grasshopper, Karamba and Kangaroo will help in designing architecturally appealing bridge solutions. Material properties will depend on the requirements (economical), but since each project will be individual, the material properties, infill geometry, and shape of the structure will vary. Future bridge design should be unlimited in all its segments, from modeling and design to its production.

**Author Contributions:** Conceptualization, L.S., A.S., J.G. and D.D.; methodology, L.S. and A.S.; software, L.S.; validation, L.S., A.S., J.G. and D.D.; formal analysis, L.S.; investigation, L.S.; resources, L.S., A.S., J.G. and D.D.; data curation, L.S., A.S., J.G. and D.D.; writing—original draft preparation, L.S. and A.S.; writing—review and editing, L.S., A.S., J.G. and D.D.; visualization, L.S.; supervision, A.S., J.G. and D.D.; project administration, J.G. and D.D.; funding acquisition, A.S., J.G. and D.D. All authors have read and agreed to the published version of the manuscript.

**Funding:** This research received no external funding.

**Institutional Review Board Statement:** Not applicable.

**Informed Consent Statement:** Not applicable.

**Data Availability Statement:** Data available on request due to restrictions, e.g., privacy or ethical. The data presented in this study are available on request from the corresponding author.

**Conflicts of Interest:** The authors declare no conflict of interest.

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
