# Peer review of "FRP Pedestrian Bridges—Analysis of Different Infill Configurations"

_buildings, doi:10.3390/buildings11110564_

Round 1

Reviewer 1 Report

It would be great to add pictures in the text of each system (Hardcore, Superdeck, DuraSpan, EZSpan, Strongwell).

Page 5, lines 214-216. The stiffness of each deck analyzed depends also on the dimensions, angle of inclination, thickness of cross-sectional parts. Therefore, it would be great to have an additional Figure with exact dimensions of each deck cross-section used in FEM design.

There is no k) and l) pictures in Figure 2 despite they are described in Figure 2.

Page 7, Line 242. it is interesting why mesh elements were chosen of 50 x 50 mm. How it is justified. Does this size of mech elements is uniform even for the infill of the deck? Maybe other sizes of mesh elements would be more appropriate.

Marking of Poisson ratio is different in Table 2 and description

Usually, the accuracy of FEM model should be justified with experimental results. The accuracy of FEM design in this article could be justified at least with one experimental research results. Maybe experimental data could be found in references of this article: [12], [16] or others.

Authors could provide a broader analysis of FEM design for instance with comparison of strain and stresses.

Reviewer 2 Report

This contribution deals with “FRP pedestrian bridges – analysis of different infill configuration”. In particular, via finite element modelling, it investigates the influence of the cross section, material and shape on the maximal deflection of the bridge decks under static flexural loading. The research is useful and worth publishing after resolving the suggested corrections below:

  1. The abbreviation FRP should be defined.
  2. In Table 1, double check units for weight (or use other characteristic).
  3. In line 117, “two-dimensional” is confusing. Maybe “flat” should be used instead.
  4. Caption for Figure 1 should describe all three pictures presented there.
  5. In Figure 2, items k) and l) are described in the caption but not presented.
  6. For the material properties specified in Table 2, reference would be useful. The same symbol should be used for Poisson’s ratio.
  7. In Figures 4-9, items a) and b) should be described in the caption. Legends for displacement magnitude and locations of maximal displacements are not visible.
  8. Please describe how the simply supported boundary conditions were realized in Abaqus. According to deformed shapes (Fig. 4, item e), for instance) the deck are rather clamped than simply supported.
  9. In Figure 7, item d) is identical to item d) in Figure 8. Strongwell deck should be presented instead.
  10. In Table 3, double check units for mass (or use other characteristic).

Reviewer 3 Report

This paper investigates the effects of different infill configurations on FRP pedestrian bridges by using numerical methods. The novelty is limited, but the results are promising. The reviewer recommends publication after the following minor comments are addressed.

  1. The author is suggested to summarize the research significance with one sentence in the abstract.
  2. The author mentioned that 3D printing model can be optimized in shape, material properties, and infill cell configuration. However, the corresponding content is only a brief introduction in the article. Therefore, the abstract is suggested to remove the content about 3D printing.
  3. The conclusion is suggested to be rewritten. The reviewer believes that the last paragraph of conclusion is unnecessary, and some contents can be simplified.
  4. There are some grammatical errors in the paper. Please check it carefully.

Round 2

Reviewer 1 Report

The Authors have improved the articles. However, the resolution of Figure 2 is too low. Additionally, additional dimensions of the infill should be added (angles of inclination, thickness etc.). Therefore, the Figure should be improved.

Reviewer 2 Report

I would like to thank the authors for carefully handling the revision. However, there are still some comments to be addressed before it can be accepted for publication.

  1. In Fig. 3, item h) does not represent the Strongwell in an arch case.
  2. According to deformed shapes (Fig. 5, item e), for instance) the decks are rather clamped than simply supported. Clarifications should be given.

Round 3

Reviewer 2 Report

The manuscript can be accepted for publication.

This manuscript is a resubmission of an earlier submission. The following is a list of the peer review reports and author responses from that submission.

Round 1

Reviewer 1 Report

Unfortunately, the article does not add anything new. It looks rushed and ill-conceived. The introduction is lengthy, outdated (like most of the cited positions), and unreal. The authors write about 100,000 composite (at least partially) bridges in some countries, and at the same time write that composites are a huge novelty in the construction sector. In fact, the composites have been widely used now - this is evidenced not only by scientific articles, but also by reports in the regular press and other media.

The article does not provide a reliable summary of the current knowledge (state of art), nor does it present any theses. It is hard for the reader to draw any conclusions. Sometimes paragraphs are in contradiction to each other.

Some of most important methodological and logic issues:

1.The article tries to solve obvious ("The likelihood of quasi-isotropic material will result in higher deformation") or artificial and unreal problems ("defining material as quasi-isotropic makes the design process simpler").

2. Deformations govern the design (f.e. 134) - what was the limit? That is the most important design assumption.

3. "The maximal span of sandwich panels is 10 m." (line 140) - where does it was come from? 800 meters from my University, a FRP sandwich footbridge of more than double span is now being built and will be opened next month. As I said - data and info in article are very outdated.

4. "Structural elements are under their ultimate capacity so that the collapse of individual connection does not result in a collapse of the whole structure." - as bridge engineer and designer I think that reapeting such thesis is dangerous. It's quite obvious, that after connection failure, there will be completely now stress state in individual members. Was that taking into account? Actually, how these connections were designed and checked? No information in the article.

5. Figure 2 - "after eight years of maintenance" - I don;t think that authors mentioned maintenance. Maybe "in service"? Maybe exploatation?

6. Leftovers from manuscript template - line 148-150.

7. Line 276: "Approximately each plate system weights around 5 kg/m2". I don't think this could happen...

8. There are mistake in table numbering - two of 1s. instead of 3th.

9. Eurocomp 1390 in citation? What 1390 stands for?

10. Last (but no least), the Authors used Abaqus for calculations. Why von Mises stresses? This is not appropriate hypothesis for laminate. Especially, that you said absolutely nothing about shear strength of laminates. Description of the material (engineering parameters and strength parameters) is not sufficient).

Reviewer 2 Report

The manuscript entitled “FRP pedestrian bridges – possibilities of design and optimization” presented a short review and a numerical model to investigate the design of FRP sandwich structures. The results are interesting while corrections must be made before consideration for publication. The comments are as followings.

(1)     The title is confusion, and it is recommended to revise and sharpen the title to emphasise the significance and novelty of this study.

(2)  In abstract, “FRP bridges are not a novelty, and still, their rise is relatively slow in the construction world” is very confusion, what is the point of the authors?

     FEM should be explained when it is firstly presented.

     “In this preliminary analysis, a material with defined fiber orientation showed much better behavior than quasi-isotropic material.”, what is the specific “behavior”?

(3)     In introduction, “…consisting of 30-70% fibers or 50 % of its total weight…”, where it is contradicted between “30-70% fibers” and “50 % of its total weight”, please be consistent.

(4)  In Section 2.1, the advantages of FRP materials and sandwich structures should be strengthened by referring more articles, such as:

  1. For high specific stiffness and strength: Composite Structures, 263: 113674
  2. For overall better properties, such as sustainability, etc.: International Journal of Crashworthiness, 25(5): 517-526
  3. For design and optimisation of sandwich structures: International Journal of Mechanical Sciences, 183: 105829

(5)    In Section 2.1, there is a bold “3. Results” in the context, please carefully check and revise it and all relevant problems if applicable.

(6)  In Section 3, what are “live load of 5 kN/m2 and dead-280 load of 1 kN/m2”? What and where are these forces being exactly imposed on? Furthermore, the boundary conditions are not clear, the authors should denote them very clearly in a graph.

(7)  In simulation, more details including the meshwork, structural geometries, etc., should be specifically presented.

(8)  The properties for the infill materials of sandwich structures should be listed.

(9) In Fig. 7-10, the legends are too small to review, please magnify them and it is recommended to use white background.

(10) In Fig. 11, what is this displacement? It is measured against which point? Please depict it in the structure.

(11) In Fig. 12, please correct to “von Mises stress”, and it is confusing. Do the authors mean a sum of all von Mises stress? Or the peak/maximums von Mises stress? Please clarify.

(12) English should be well improved and the whole manuscript should be carefully checked in terms of grammar and formal expression. e.g. “FRP bridges are not a novelty, and still, their rise is relatively slow in the construction world”, etc.

Round 2

Reviewer 2 Report

Thanks for revision.